# Demonstration and uncertainty analysis of synchronised scanning lidar measurements of 2D velocity fields in a boundary-layer wind tunnel

Marijn Floris van Dooren[1], Filippo Campagnolo[2], Mikael Sjöholm[3], Nikolas Angelou[3], Torben Mikkelsen[3], and Martin Kühn[1]

[1]ForWind, University of Oldenburg, Institute of Physics, Oldenburg, Germany
[2]Wind Energy Institute, Technical University of Munich, Garching, Germany
[3]Dept. of Wind Energy, Technical University of Denmark, Roskilde, Denmark

*Correspondence to:* Marijn Floris van Dooren (marijn.vandooren@uni-oldenburg.de)

**Abstract.** This paper combines the research methodologies of scaled wind turbine model experiments in wind tunnels with short-range WindScanner lidar measurement technology. The wind tunnel of the Politecnico di Milano was equipped with three wind turbine models and two short-range WindScanner lidars to demonstrate the benefits of synchronised scanning lidars in such experimental surroundings for the first time. The dual-lidar system can provide fully synchronised trajectory scans with sampling time scales ranging from seconds to minutes. First, staring mode measurements were compared to hot-wire probe measurements commonly used in wind tunnels. This yielded goodness of fit coefficients of 0.969 and 0.902 for the 1 Hz averaged $u$- and $v$-components of the wind speed, respectively, validating the 2D measurement capability of the lidar scanners. Subsequently, the measurement of wake profiles on a line as well as wake area scans were executed to illustrate the applicability of lidar scanning to the measurement of small scale wind flow effects. An extensive uncertainty analysis was executed to assess the accuracy of the method. The downsides of lidar with respect to the hot-wire probes are the larger measurement probe volume, which compromises the ability to measure turbulence, and the possible loss of a small part of the measurements due to hard target beam reflection. In contrast, the benefits are the high flexibility in conducting both point measurements and area scanning, and the fact that remote sensing techniques do not disturb the flow while measuring. The research campaign revealed a high potential for using short-range synchronised scanning lidars to measure the flow around wind turbines in a wind tunnel, and increased the knowledge about the corresponding uncertainties.

## 1 Introduction

During the past few years, several research groups have focused attention on wind tunnel experiments with the innovative idea of supporting research not only related to the validation of purely aerodynamic models, but mainly to support numerical activities on control and aero-servo-elasticity (Bottasso et al., 2014) as well as understanding the interaction of wind turbines with turbulent flow (Rockel et al., 2014). In fact, testing of wind turbines in full-scale in the atmospheric boundary-layer imposes several constraints, such as the difficulty in having an accurate knowledge and repeatability of the environmental

conditions, higher costs, and especially for public researchers, the difficulty to have access to industrial wind turbines as a research platform. In the same period, academic and industrial researchers have developed new scanning wind lidars able to map full three-dimensional vector wind and turbulence fields in 3D space (Mikkelsen, 2012; Wagner et al., 2015; Simley et al., 2016). Even for complex flows, such as the flow around wind turbines, the lidars can be applied without disturbing the

5 flow itself. The present work reports on the testing activity recently conducted by a joint team of research groups, where two short-range WindScanners have been used in a boundary-layer test section of a wind tunnel for the first time, in order to map the flow of the free chamber as well as to accurately measure the wakes of scaled wind turbine models. Previous research has already been done on wake analysis of full-scale turbines (Iungo and Porté-Agel, 2014; Banta et al., 2015) and in wind tunnels (Lignarolo et al., 2014; Iungo, 2016). Please note that one of the shortcomings of measuring in a wind tunnel with respect to

10 free-field measurements is the poor ability of simulating the variability of atmospheric stability and a representative wind rose.

The objective of the research presented in this paper is to assess the capabilities of continuous-wave short-range lidar to map wind flows and measure turbulence in a wind tunnel. The study has been executed within the larger scope of the CompactWind project, which has the purpose of investigating the effect of different wind farm control concepts and yaw configurations of the individual turbines on the wind farm energy output, the wake structures and wind turbine loads (Campagnolo et al., 2016c).

In section 2 about the methodology, the wind tunnel, the model wind turbines, the lidars and the hot-wire probe are described and afterwards the three sequential experiments executed during the measurement campaign are introduced. The results of each of these three experiments are then presented and discussed in section 3, together with an extensive uncertainty analysis. Finally the paper is concluded in section 4.

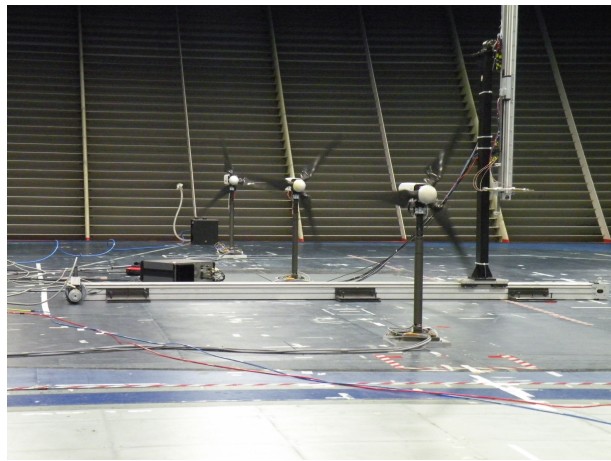

**Figure 1.** Model wind turbines in operation in the wind tunnel.

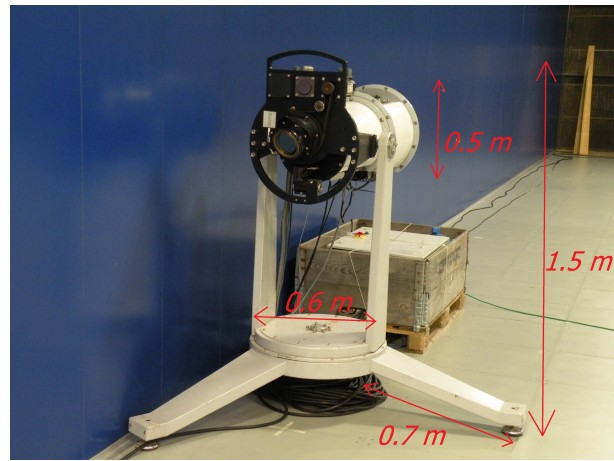

**Figure 2.** One of the two WindScanner lidars inside the wind tunnel.

## 20 2 Methodology

The experimental setup in the wind tunnel (Bottasso et al., 2014) of the Politecnico di Milano (PoliMi) consisted of three generically scaled wind turbine models (see Fig. 1), named *G1*, specifically designed by the Wind Energy Institute (WEI) at

the Technical University of Munich (TUM) for wind farm control research applications (Campagnolo et al., 2016a, b, c), as well as two short-range WindScanners (see Fig. 2) developed by the Department of Wind Energy of the Technical University of Denmark (DTU), joining the common measurement campaign during the last week of January 2016.

## 2.1 The wind tunnel facility

The PoliMi wind tunnel has a closed-return configuration facility arranged in a vertical layout with two test sections. The boundary-layer test section, sketched in Fig. 3, is located at the upper level in the return duct and has a cross-sectional area of 13.84 m by 3.84 m and a length of 36 m, illustrated by the blue outer boundaries. The three wind turbines were mounted on a turn table which allows for rotating the entire turbine array setup, as to create a lateral offset between the wind turbines. When the turn table is in its 'home position', the turbines line up in $x$-direction with a distance of $4D$ between them. The

WindScanners are indicated with red rectangles and their commanded synchronised scan pattern for scanning the wind turbine wakes is plotted in grey. Typical vertical profiles of wind speed and turbulence can be imitated by the use of bricks on the floor that act as roughness and turbulence generators, i.e. spires, placed at the chamber inlet at the left boundary. For more information about the wind tunnel, please refer to Zasso et al. (2005).

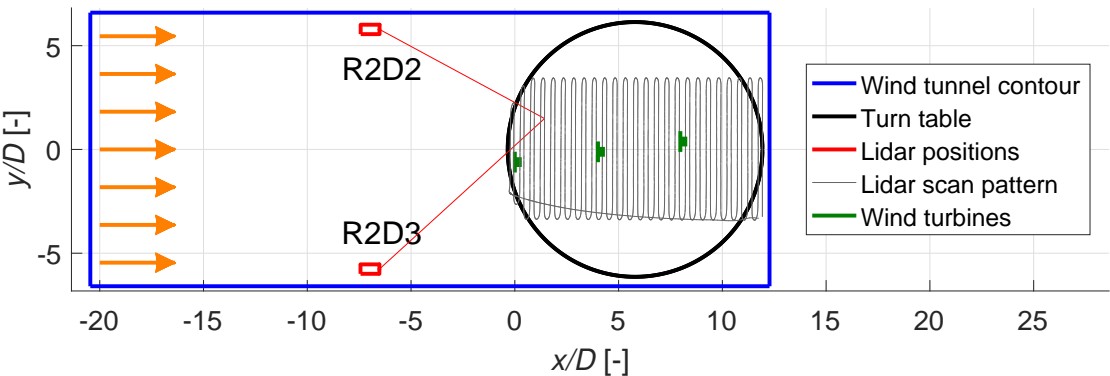

**Figure 3.** Configuration of the wind tunnel of the Politecnico di Milano with the two scanning lidars and the three model wind turbines installed on the turn table. The axes are normalised with respect to the wind turbine diameter of $D = 1.1$ m.

The inflow conditions in the wind tunnel were kept constant throughout the measurement campaign. The free-stream wind

speed and the turbulence intensity, both at hub height, were $u_\infty = 5.67$ m s$^{-1}$ and $TI = 5.5\%$, respectively. Figures 4 and 5 illustrate vertical profiles of the inflow wind speed and turbulence intensity, respectively, measured during a highly similar measurement campaign. The vertical wind profile corresponds to a power law profile with shear exponent $\alpha = 0.09$.

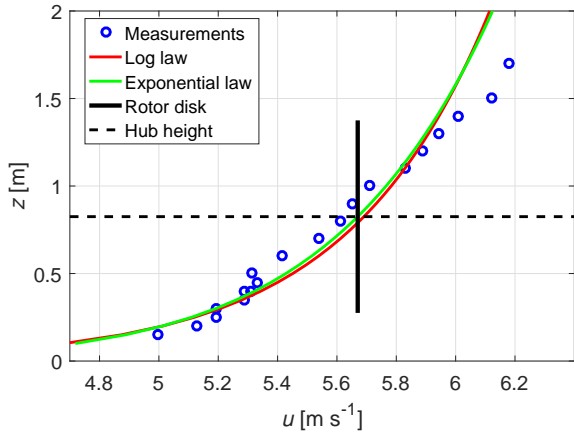

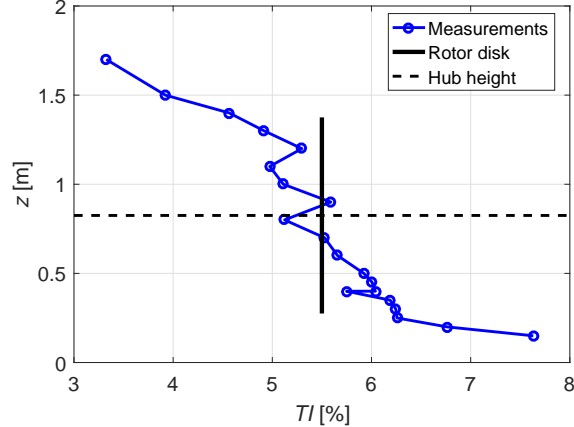

**Figure 4.** Vertical profile of the inflow velocity.

**Figure 5.** Vertical profile of the inflow turbulence intensity.

## 2.2 The G1 wind turbine models

The three scaled *G1* wind turbine models have a rotor diameter $D$ of 1.1 m, a hub height of 0.825 m and a rated rotor speed equal to 850 rpm. They were designed to provide a realistic energy conversion process, which means reasonable aerodynamic
loads and damping with respect to full-scale wind turbines, as well as wakes with a realistic geometry, velocity deficit and turbulence intensity. Moreover, systems have been integrated to enable individual blade pitch, torque and yaw control, while a sufficient onboard sensor equipment of the machine, providing measurements of rotor azimuth angle, main shaft loads, rotor speed and tower base loads, enables the testing of both wind turbine and wind farm control strategies.

Each *G1* model is equipped with three blades whose pitch angles can be varied by means of brushed motors housed in the
hollow roots of the blades and commanded by dedicated electronic control boards housed in the hub spinner. Electrical signals from and to the pitch control boards are transmitted by a through-bore 12-channels slip ring located within the rectangular carrying box holding the main shaft. A torque sensor allows for the measurement of the torque provided by a brushless motor located in the rear part of the nacelle, which is operated as a generator by using a servo-controller. An optical encoder, located between the slip ring and the rear shaft bearing, allows for the measurement of the rotor azimuth angle. The tower is softened
at its base by machining four small bridges, on which strain gauges are glued so as to measure fore-aft and side-side bending moments. Aerodynamic covers of the nacelle and hub ensure a satisfactory quality of the flow in the central rotor area.

Each *G1* model is controlled by an *M1 Bachmann* hardware real-time module. Similarly to what is done on real wind turbines, collective or individual pitch-torque control laws are implemented on and real-time executed by the control hardware. Sensor readings are used online to properly compute the desired pitch and torque demands, which are in turn sent to the actuator
control boards via analog or digital communication.

## 2.3 The short-range WindScanner lidars

The two short-range WindScanners R2D2 and R2D3, installed near the section walls upwind of the turbine models (see Fig. 3), are continuous-wave, coherent lidars that can provide Doppler spectrum averaged wind speeds at rates up to 390 Hz. The measurement range is defined by the optical configuration of the device, which enables motor controlled focus distance between about 10 m and 150 m. The longitudinal line-of-sight sampling volumes can become very small at short ranges, e.g. about 13 cm probe length at a 10 m focus distance, thus the WindScanners were placed as close as possible to the measurement area of interest, within the reachable focus distances. The laser beam can be freely steered within a cone with a full opening angle of 120° by the use of two prisms. The two prism motors and focus motor that each lidar comprises are synchronously operated by a common central multi-axis motion controller that steers all the six motors such that the two focused laser beams can synchronously follow a common scanning trajectory. The relation between the motor positions and the measurement location relative to each WindScanner was pre-calibrated at DTU leaving only the location and orientation of the WindScanners relative to the measurement scene to be determined in the deployment situation. The scan heads of R2D2 and R2D3 were placed at $(x = -7.17 \text{ m}, y = 6.36 \text{ m}, z = 1.31 \text{ m})$ and $(x = -7.18 \text{ m}, y = -6.34 \text{ m}, z = 1.30 \text{ m})$, respectively, and the instruments were tilted by 90.148° and 90.123°, respectively. The symmetry axes of the scan heads were roughly aligned with the $x$-axis, i.e. with azimuth directions relative the $x$-axis of -0.111° and 0.021°, respectively. The detailed positions where obtained by a Leica total station and the orientation was obtained by the similarly determined detailed positions of small rotating balls placed close to the wind tunnel outlet and providing distinct hard target Doppler returns. An additional verification of the measurement locations close to the turbines was done by imaging the laser beam on a reflective plane by an infrared-sensitive camera.

Each lidar measures a projected line-of-sight component of the three-dimensional wind velocity vector. From two temporally and spatially synchronised line-of-sight measurements $v_{LOS}$, the $u$- and $v$-components of the wind speed, defined along and lateral to the main wind direction respectively, can be calculated by solving the linear equation system in Eq. (1):

$$
\begin{bmatrix} \cos(\chi_1)\cos(\delta_1) & \sin(\chi_1)\cos(\delta_1) \\ \cos(\chi_2)\cos(\delta_2) & \sin(\chi_2)\cos(\delta_2) \end{bmatrix} \begin{bmatrix} u \\ v \end{bmatrix} = \begin{bmatrix} v_{LOS_1} \\ v_{LOS_2} \end{bmatrix}
\tag{1}
$$

The horizontal and vertical scanning angles of a lidar system are the azimuth $\chi$ and elevation $\delta$ angles, respectively. The vertical wind component $w$ is omitted, since a third lidar would be needed to evaluate this additional component. Because the lidar scan heads are located slightly higher than the turbine hub height, small negative elevation angles of up to $\delta < 3°$ had to be used. This could create a bias of $\sin(3°)w$ on the measured $v_{LOS}$.

As mentioned before, the lidars acquire each measurement based on the aerosols present in a certain probe volume, which is illustrated qualitatively in Fig. 6. The value of the probe length is commonly defined as twice the Half Width Half Maximum $\Gamma$, which is the distance at either side of the focus point at which the backscatter signal power is reduced to half of its maximum power. The power spectrum of the backscattered signal can be expressed with a Lorentzian probability distribution along the beam line-of-sight direction, multiplied by the line-of-sight wind speed component at the corresponding coordinates. The probe length increases quadratically with the focus distance, which is expressed in Eq. (2) and plotted in Fig. 7. In Eq. (2), $\Gamma$ is the

Half Width Half Maximum, $\lambda$ is the lidar laser wavelength (1.565 $\mu$m, infrafred), $f$ is the focus distance and $a$ is the laser beam width at the aperture (28 mm). Throughout the measurement campaign focus distances between 10 m and 20 m were used, yielding probe lenghts of 13 cm and 50 cm, respectively. These marks are indicated in Fig. 7.

$$\Gamma = \frac{\lambda f^2}{\pi a^2} \tag{2}$$

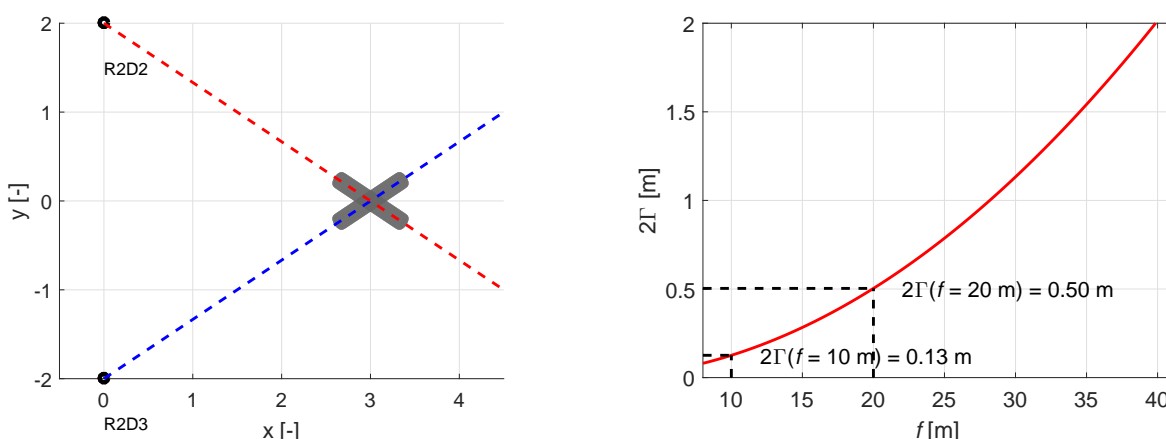

**Figure 6.** Sketch of the probe volumes of both lidars (not to scale). **Figure 7.** Relationship between focus distance and the probe length.

### 2.4 The hot-wire probe

A tri-axial Dantec 55R91 hot-wire probe (see Fig. 8) was mounted on an automatic traversing system (see Fig. 9) and provided 2500 Hz measurements of the three-dimensional wind speed vector in the wind tunnel. The three wires of the hot-wire probe form an orthogonal system with respect to each other and are also positioned orthogonally to the prongs of the probe for increased accuracy. The effective sensor length of each of the wires is 1.25 mm. The calibration of the hot-wire probes was performed in the 150-by-200 mm$^2$ closed test section within the PoliMi wind tunnel facility, with a contraction ratio of 25. The calibration procedure consists of the following three steps:

1. The probe in positioned in the wind tunnel, whose flow velocity is calculated from the dynamic pressure, measured with a pressure transducer of the type *Druck LPM9481*, and the density, in turn calculated using accurate measurements of the relative humidity and absolute pressure.

2. The alignment of the probe support to the wind tunnel flow is assured by means of an inclinometer of the type *Spectron L-212T*.

3. The Jørgensen (2002) Law is applied, which follows the hypothesis of decoupled directional response and velocity magnitude response of the probe.

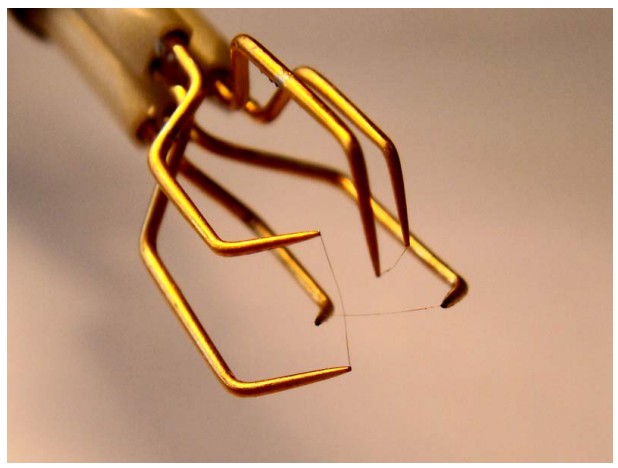

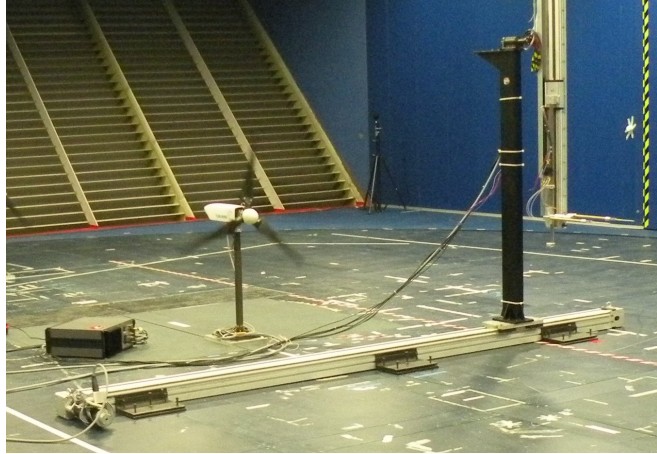

**Figure 8.** The tri-axial Dantec 55R91 hot-wire probe.    **Figure 9.** The traversing system for the hot-wire probes.

## 2.5 Measurement examples

The measurement campaign covered several scenarios. In this paper, only relevant examples from the three main types of measurements are presented to illustrate the capabilities:

1. *Comparison between lidar and hot-wire probe measurements*: The lidar beams were focused as closely to the hot-wire as practically possible, i.e. without influencing the hot-wire probe due to heating by the laser beams on the one hand, or blocking the view of the lidars with the hot-wire probe and the traversing system on the other. The focus offset was chosen to be 2 cm. A series of different points in the wind tunnel were measured by both anemometers for a duration of 2 minutes each. In this case, the data of a single point at $x = 2.23$ m, $y = 0.88$ m, $z = 0.83$ m is considered for analysis.

The wind turbines were idling at approximately 80 rpm, which is assumed to have a negligible effect on the flow.

2. *Measurement of wake profiles along a horizontal line*: The lidars performed measurements back and forth along cross-wind lines at several distances downstream of the first wind turbine at hub height and spanning $\pm 3.5D$ around the wake centre. The complete line was covered every 1 s with equally sampled measurements. In the case presented, a wake profile at a $3D$ downstream distance of the first wind turbine is analysed. This turbine was operating with an average

rotor speed of 805 rpm, average pitch angle of $0.4°$, $C_p$ of 0.38, $C_T$ of 0.83 and had a yaw offset of $20°$.

3. *Measurement of horizontal wake area scans*: The full area containing the three wakes of the model turbines was mapped by the lidars by iterating through the scanning pattern indicated previously in Fig. 3. The scans cover an area of 7 m by 13 m every 18.5 s. Multiple scans were averaged to resolve the mean wake features. None of the wind turbines had a yaw offset. The first turbine had an average rotor speed of 830 rpm, average pitch angle of $0.55°$, $C_p$ of 0.42 and $C_T$ of 0.88.

The second and third turbine had average rotor speeds of 710 and 736 rpm, respectively. No $C_p$ and $C_T$ were recorded.

## 3 Results

### 3.1 Comparison between lidar and hot-wire probe measurements

The first step of the lidar campaign in the wind tunnel was to establish a quantitative measure of the accuracy of the lidars with respect to the commonly applied devices in such an environment, i.e. hot-wire anemometers. Here, the established hot-wire probe served as a validation for the lidar measurements. In order to compare the devices directly with each other, the hot-wire probe data recorded at 2500 Hz has been averaged to match the lidar measurement frequency of 390 Hz. Subsequently, the data of both devices was averaged to 1 Hz and compared again.

**Table 1.** Comparison of the statistics of the 390 Hz and 1 Hz wind speed components measured at a point over a 2-minute time frame by the hot-wire probe and the lidars.

| | Hot-wire 390 Hz | | | Lidar 390 Hz | | Hot-wire 1 Hz | | | Lidar 1 Hz | |
|---|---|---|---|---|---|---|---|---|---|---|
| | $u$ | $v$ | $w$ | $u$ | $v$ | $u$ | $v$ | $w$ | $u$ | $v$ |
| $\mu$ [m s$^{-1}$] | 5.67 | -0.04 | 0.08 | 5.65 | -0.03 | | | | | |
| $\sigma$ [m s$^{-1}$] | 0.31 | 0.28 | 0.26 | 0.28 | 0.27 | 0.15 | 0.07 | 0.08 | 0.16 | 0.07 |
| $\gamma_1$ [-] | -0.05 | -0.16 | -0.03 | -0.21 | 0.67 | -0.09 | -0.24 | 0.02 | -0.13 | -0.29 |
| $\gamma_2$ [-] | 3.17 | 3.03 | 3.03 | 4.55 | 15.13 | 3.14 | 3.02 | 2.69 | 3.03 | 3.20 |

In Table 1 the mean ($\mu$), standard deviation ($\sigma$), skewness ($\gamma_1$) and kurtosis ($\gamma_2$) of the $u$-, $v$-, and $w$-components from a single 2-minute point measurement time series of both systems can be seen, for the 390 Hz time series and the 1 Hz averaged time series. The $u$-, $v$- and $w$-components are expressed in the $x$-, $y$- and $z$-direction of the lidar reference frame, respectively, which is indicated in the wind tunnel configuration sketch (see Fig. 3). The hot-wire probe measurements originally obtained in a different coordinate system were transformed into the lidar frame of reference. Time synchronisation between the devices was taken care of by a cross-correlation optimisation procedure.

The $u$-, $v$- and $w$-components of the hot-wire are directly measured and the $u$- and $v$-components of the lidars are derived from the line-of-sight measurements by applying Eq. (1). It cannot be confirmed with certainty whether the nonzero $v$- and $w$-components are a flow feature or stem from an instrument misalignment. The $w$-component cannot be evaluated from the lidar measurements in this case and is therefore neglected. Doing this may cause a slight bias on the $u$- and $v$-components measured by the lidars, since on average there is a vertical wind speed of 0.08 m s$^{-1}$. It was mentioned before that the measured $v_{LOS}$ will be affected by a bias of $\sin(3°)w$, which gets propagated through Eq. 1.

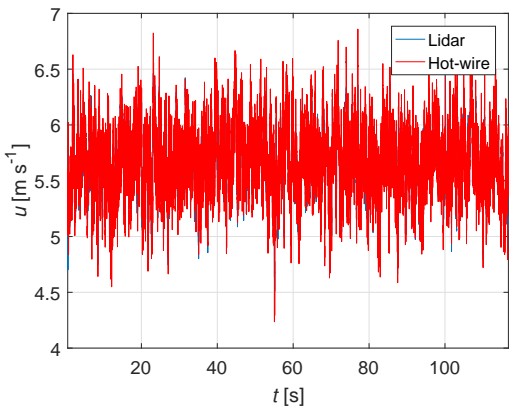

**Figure 10.** Visual comparison of the 390 Hz $u$-component.

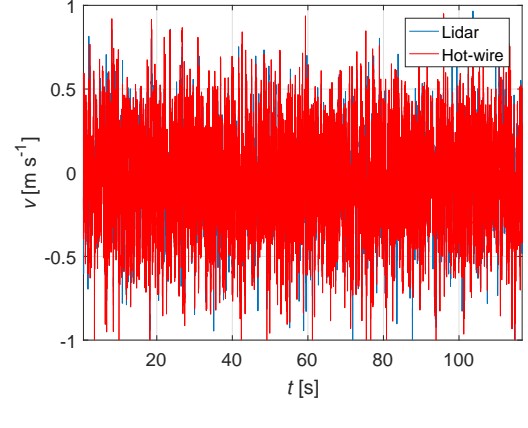

**Figure 11.** Visual comparison of the 390 Hz $v$-component.

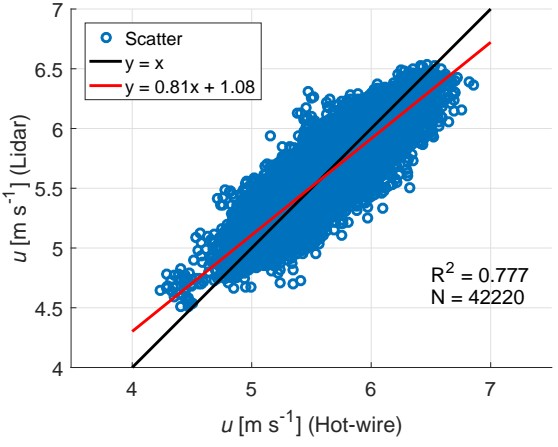

**Figure 12.** Correlation of the 390 Hz $u$-component.

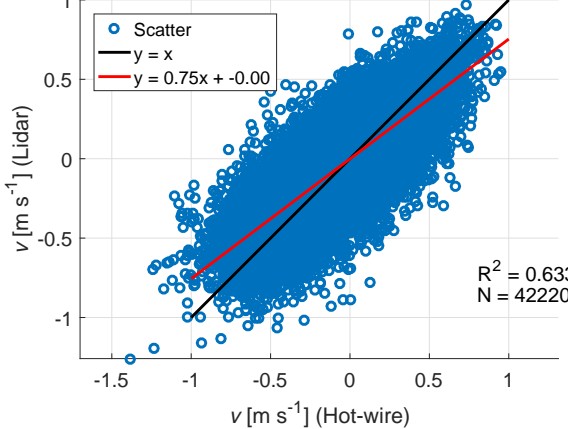

**Figure 13.** Correlation of the 390 Hz $v$-component.

Time series of the 390 Hz $u$- and $v$-components of both the lidar and the hot-wire probe can be seen in Figs. 10 and 11, respectively. The signals are hard to distinguish from each other, because of the good correlation. Correlation plots of both the $u$- and $v$-components are shown in Figs. 12 and 13, respectively. Although the regression line does not perfectly resemble $x = y$ and some scattering is visible, the measurements yielded very reasonable - especially for the considered sampling rate - goodness of fit coefficients of $R^2 = 0.777$ for the $u$-component and $R^2 = 0.633$ for the $v$-component. A possible reason for the remaining scatter in the plot is the difference in the probe volumes of the anemometers. They are not measuring in the exact same point or volume, so different fluctuations are seen by the different devices. The biases in the slope and the offset could be caused by neglecting the contribution of the $w$-component on the measured $v_{LOS}$. Also there might be a small bias in the transformation between the different coordinate systems of the lidars and the hot-wire probe, causing a cross-contamination in the calculation of both wind speed components $u$ and $v$.

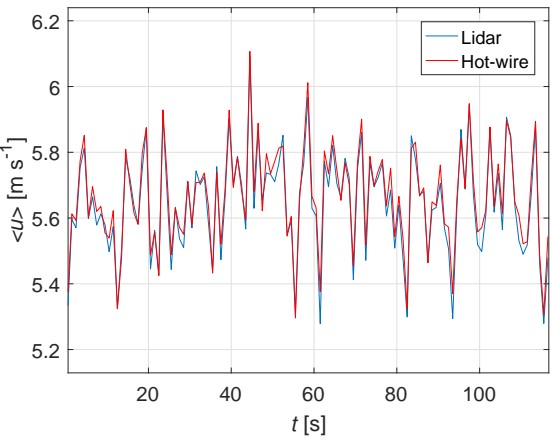 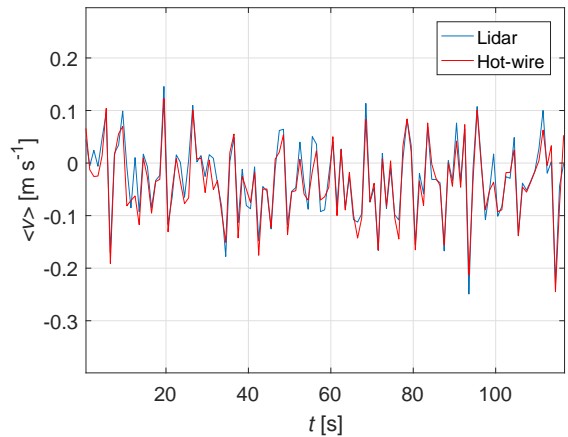

**Figure 14.** Visual comparison of the 1 Hz averaged $u$-component. **Figure 15.** Visual comparison of the 1 Hz averaged $v$-component.

The time series were subsequently further averaged to 1 Hz data. Figures 14 and 15 display the $u$- and $v$-components of both anemometers, respectively. On this time scale, it can be concluded visually that the measurements correlate very well. The 1 Hz averaged $u$- and $v$-components in Figs. 14 and 15 were correlated with each other, as shown by Figs. 16 and 17.

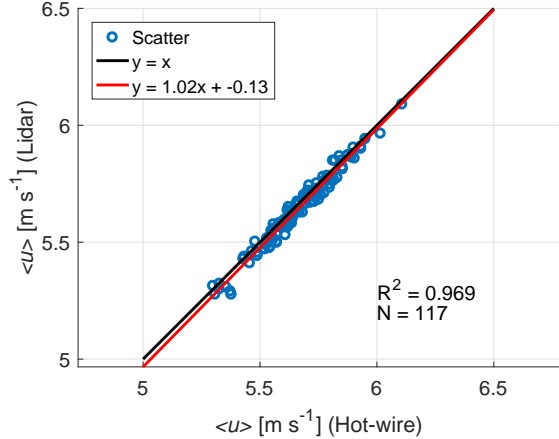 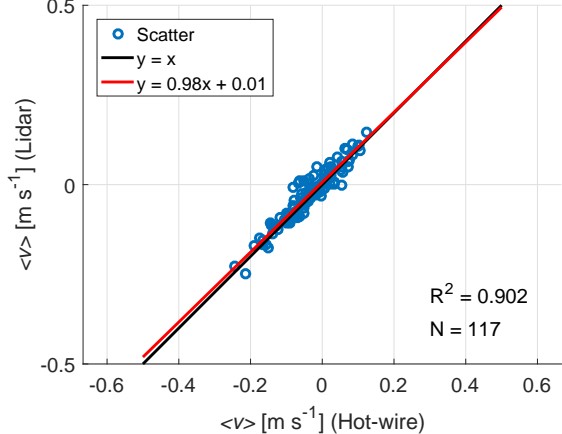

**Figure 16.** Correlation of the 1 Hz averaged $u$-component. **Figure 17.** Correlation of the 1 Hz averaged $v$-component.

The mentioned effects that caused the scatter in the correlated 390 Hz data of Figs. 12 and 13 do not play a large role anymore after the data has been averaged to 1 Hz time series, since small scale fluctuations are averaged out. Correlating the 1 Hz averaged data now provided the goodness of fit coefficients of $R^2 = 0.969$ for the $u$-component and $R^2 = 0.902$ for the $v$-component, which can be regarded as a definite validation of the lidar measurements in the wind tunnel at 1 Hz. The fact that

10    both components at 1 Hz follow the same trend, is a confirmation of the good synchronisation of the WindScanners.

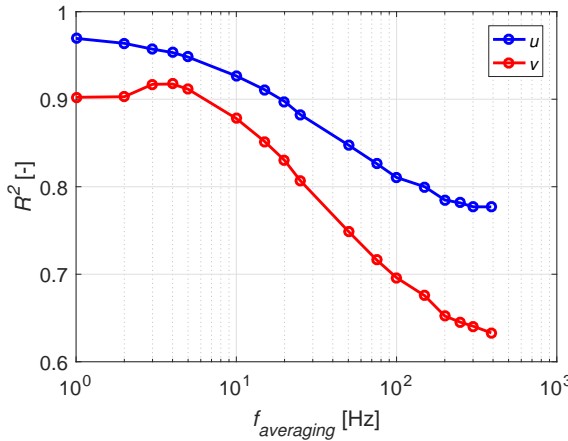
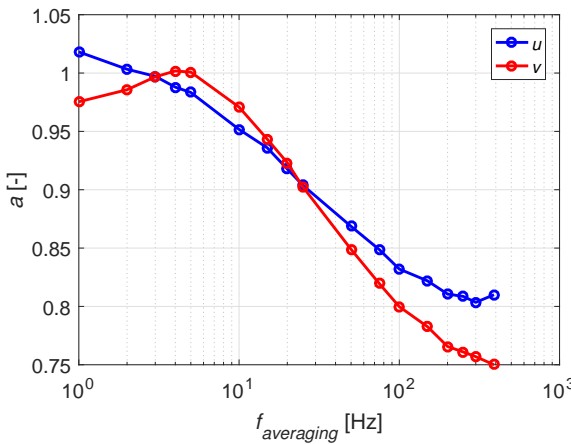

**Figure 18.** Goodness of fit parameter $R^2$ of the linear fit between the averaged $u$- and $v$-components of the lidar and the hot-wire as a function of averaging frequency.

**Figure 19.** Slope $a$ of the linear fit between the averaged $u$- and $v$-components of the lidar and the hot-wire as a function of averaging frequency.

Figures 18 and 19 show the influence of the averaging frequency on the goodness of fit $R^2$ and the regression line slope $a$ of the linear fit, respectively. As expected, a better fit is yielded for lower averaging frequencies. In case of wake measurements, it is important that the lidars are able to resolve the fluctuation scales induced by the wind turbines. Since the rated rotor speed is equal to 850 rpm, which is approximately 14 Hz, also the time series with this averaging rate were compared. The goodness of fit coefficients were $R^2 = 0.916$ for the $u$-component and $R^2 = 0.860$ for the $v$-component in this case.

To analyse the capability of measuring turbulence with lidars (Sathe and Mann, 2013), the spectra of the $u$-component of both the lidar and the hot-wire are plotted in Fig. 20. The spectra are based on the full two minute time series of the 390 Hz data, which is split into ten blocks, filtered with a Hann window for smoothing, and then averaged. Since the sampling frequencies of the hot-wire probe and the lidars are 2500 Hz and 390 Hz, respectively, the boundary of the plot was chosen to be the Nyquist frequency based on the lidar, which equals 195 Hz. The lidar measurements are based on the backscatter of aerosols in a small measurement volume with a length of approximately 13 cm and therefore turbulent structures with a size smaller than this measurement probe volume are partly filtered out. By applying Taylor's Theorem (Taylor, 1938) one can calculate that the lidars can resolve temporal turbulence scales up to $\frac{1}{2}$·5.67 m s$^{-1}$/0.13 m = 22 Hz in this case. This line is marked in Fig. 20. It can be seen that the lidar indeed shows less power in the spectrum than the hot-wire for the upper frequency range. The drop in the slope of the spectrum does not exactly coincide with the 22 Hz frequency mark, because the intrinsic Lorentzian spatial weighting function of a continuous-wave lidar extends beyond the defined bounds of the probe length, therefore also acting as a filter on lower frequencies. The effect of spatial weighting is explained in detail by Sjöholm et al. (2009). Also combining measurements from two lidars that each have a different probe volume causes an even larger effect of averaging out small turbulence scales over a more complex x-shaped volume (see Fig. 6).

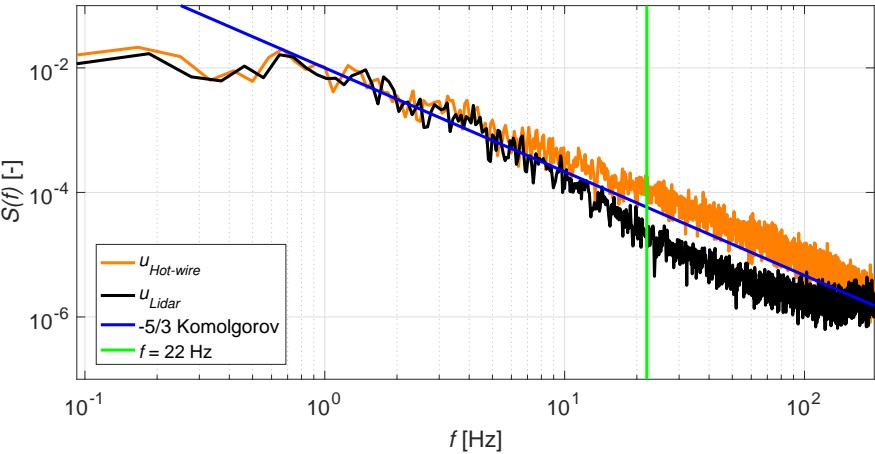

**Figure 20.** Turbulence spectrum of the $u$-component.

## 3.2 Measurement of wake profiles along a horizontal line

In Figs. 21 and 22, the respective $u$- and $v$-components of the wind speed evaluated from the line-of-sight measurements of both lidars of the transverse wake profile at hub height at a distance of $3D$ downstream of the first turbine can be seen. Both components are normalised with respect to the free-stream velocity $u_\infty = 5.67$ m s$^{-1}$. The data availability was 87.9%, due to the hard target signal return of the wind turbine blades. All single measurements recorded with 390 Hz over a 1-minute period are plotted, as well as a bin averaged line with its standard deviation ($\pm 1\sigma$) bounds. The scatter of the measurements is reasonable and a smooth wake profile is produced. It is interesting to note that the $v$-component is almost zero on average, but it has a highly turbulent behaviour at the wake boundaries. This is probably caused by the high velocity gradients and the increased turbulence intensity at the boundaries of the wake.

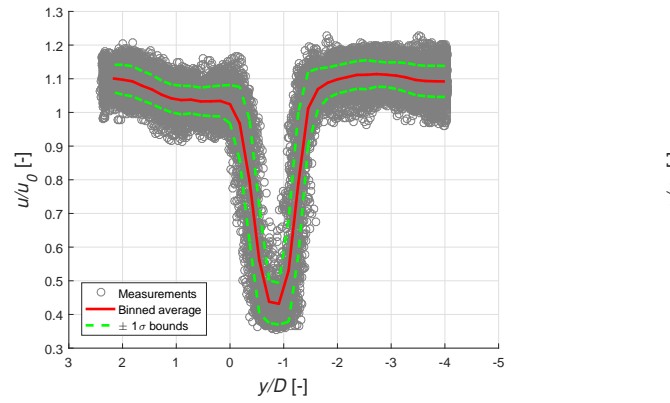

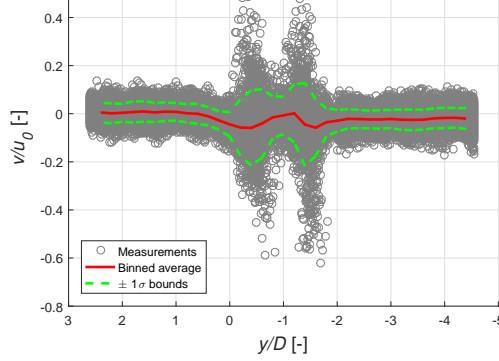

**Figure 21.** Wake profile expressed in the evaluated $u$-component, at $3D$ behind the first turbine with a $20°$ yaw angle.

**Figure 22.** Wake profile expressed in the evaluated $v$-component, at $3D$ behind the first turbine with a $20°$ yaw angle.

## 3.3 Measurement of horizontal wake area scans

The WindScanners can follow synchronised scan patterns that cover any desired plane or volume in space. The scanning pattern sketched in Fig. 3 was used to map a horizontal plane at hub height containing the wake of all three wind turbines. Note that at the far end of the scan, the lidar units are measuring at a focus distance of about 20 m, which results in a probe length of about 50 cm locally. In Fig. 23 the normalised line-of-sight component measured by R2D3 is plotted, as the result of one scan iteration (Fig. 23a) and as an average of 30 scan iterations (Fig. 23b). This amount of iterations corresponds approximately to a 10-minute period. Although some blocking of the data is expected from the moving wind turbine blades, still the measurement availability after filtering of 89.4% is satisfying. The normalised $u$- and $v$-components, calculated through Eq. (1), are plotted in Figs. 24a and 24b, respectively. It illustrates that the lidars are capable of determining the two-dimensional flow across a horizontal wind turbine wake cross-section. The plot of the $u$-components shows a smooth and overlapping triple wake, enabled by the low turbulence in the wind tunnel. The non-zero local $v$-components are indicating the initial flow expansion in the induction zone of each of the turbines. These effects are well visible in the upper part of the plot, whereas in the lower part of the wake these effects are averaged out due to the larger turbulence in the region where the wakes from the three turbines partly overlap. Some artefacts can be seen in the background of Fig. 24b, probably caused by interpolation.

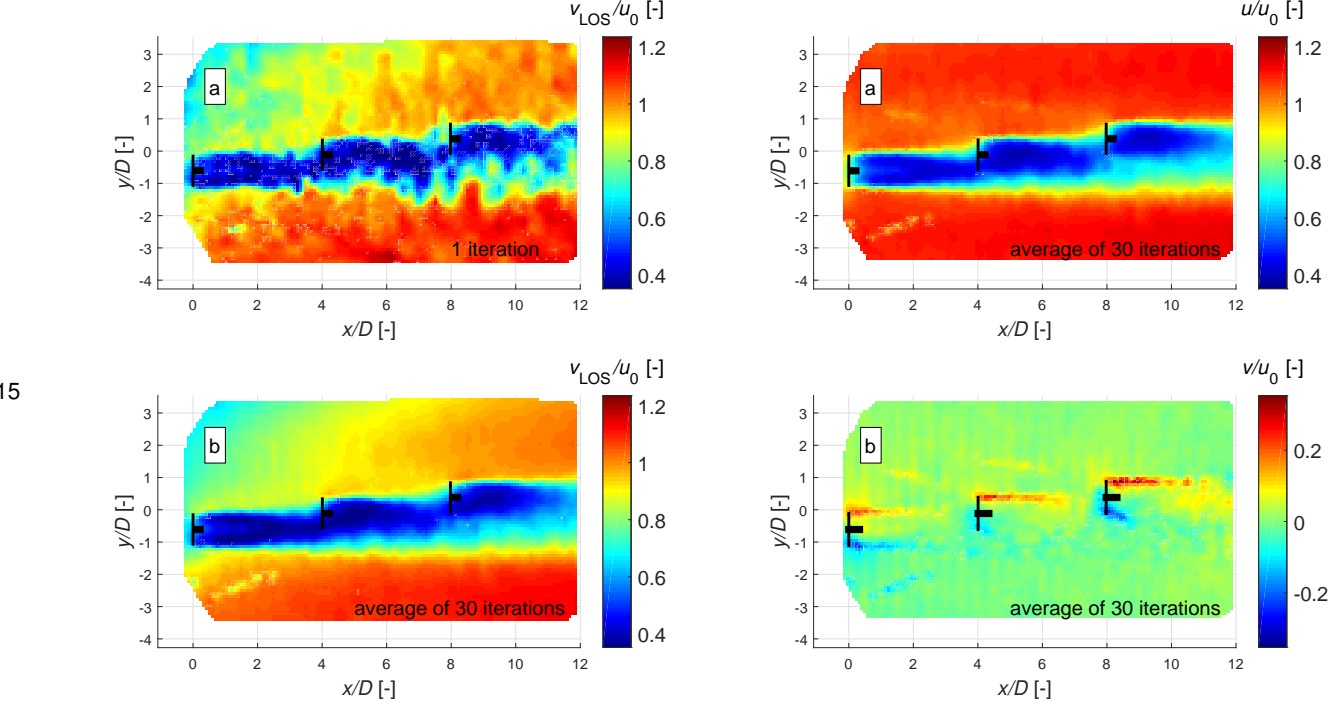

**Figure 23.** Wake wind field expressed in the line-of-sight component of R2D3, for **a**: 1 iteration, **b**: average of 30 iterations.

**Figure 24.** Wake wind field expressed in the evaluated $u$- and $v$-components; **a**: $u$, **b**: $v$.

Note that it is hard to give any hard conclusions on the lidars' ability to measure small-scale turbulent fluctuations for this case, since the temporal resolution of the scans is not sufficient to track them. The wind field 'refreshing time' is in the order of a few seconds, whereas a single scan took 18.5 s to complete. This is a direct consequence of the small scale of the experiment and the trade-off that was made between spatial and temporal resolution.

## 3.4 Uncertainty analysis

It is of particular interest how the dual-Doppler reconstruction affects the uncertainty of the estimated $u$- and $v$-components, according to a standard uncertainty propagation method (JCGM, 2008). Similar analyses have already been carried out by Stawiarski et al. (2013) and van Dooren et al. (2016). The method described here considers both the lidar measurement uncertainty itself, as well as the artificially added uncertainty of the dual-Doppler reconstruction. The inputs are:

1. The uncertainty of the measured line-of-sight wind speed $e_{v_{LOS}}$, conservatively assumed to be 1% (Pedersen et al., 2012) of the free-stream velocity (= 0.0567 m s$^{-1}$), plus the worst-case bias of $\sin(3°)w$ (= 0.0523 m s$^{-1}$) on the measured $v_{LOS}$ that could be caused by neglecting a maximum $w$-component of 1 m s$^{-1}$ (as measured by the hot-wire probe). This is a conservative estimate, which makes sure all possible error sources are included.

2. The pointing error for both the elevation and azimuth angles, $e_\delta$ and $e_\chi$ respectively, assumed to be 0.5 mrad ($\approx 0.03°$).

By solving the linear system in Eq. (1), one can express the quantities $u$ and $v$ individually:

$$u = \frac{\sin(\chi_2)\cos(\delta_2)v_{LOS_1} - \sin(\chi_1)\cos(\delta_1)v_{LOS_2}}{\cos(\delta_1)\cos(\delta_2)\sin(\chi_2 - \chi_1)} \tag{3}$$

$$v = \frac{\cos(\chi_1)\cos(\delta_1)v_{LOS_2} - \cos(\chi_2)\cos(\delta_2)v_{LOS_1}}{\cos(\delta_1)\cos(\delta_2)\sin(\chi_2 - \chi_1)} \tag{4}$$

The numerical errors $e_u$ and $e_v$ of the respective velocity components $u$ and $v$ are then expressed as follows:

$$e_u = \sqrt{\left(\frac{\partial u}{\partial v_{LOS_1}}e_{v_{LOS_1}}\right)^2 + \left(\frac{\partial u}{\partial v_{LOS_2}}e_{v_{LOS_2}}\right)^2 + \left(\frac{\partial u}{\partial \chi_1}e_{\chi_1}\right)^2 + \left(\frac{\partial u}{\partial \chi_2}e_{\chi_2}\right)^2 + \left(\frac{\partial u}{\partial \delta_1}e_{\delta_1}\right)^2 + \left(\frac{\partial u}{\partial \delta_2}e_{\delta_2}\right)^2} \tag{5}$$

$$e_v = \sqrt{\left(\frac{\partial v}{\partial v_{LOS_1}}e_{v_{LOS_1}}\right)^2 + \left(\frac{\partial v}{\partial v_{LOS_2}}e_{v_{LOS_2}}\right)^2 + \left(\frac{\partial v}{\partial \chi_1}e_{\chi_1}\right)^2 + \left(\frac{\partial v}{\partial \chi_2}e_{\chi_2}\right)^2 + \left(\frac{\partial v}{\partial \delta_1}e_{\delta_1}\right)^2 + \left(\frac{\partial v}{\partial \delta_2}e_{\delta_2}\right)^2} \tag{6}$$

The first two terms of the square root, i.e. containing the partial derivatives with respect to the line-of-sight speed, formed the largest contribution to both $e_u$ and $e_v$. The derivatives with respect to the scanning angles had less influence for the current measurement setup in combination with the assumed numerical values of the uncertainties.

To include the uncertainty introduced by measuring in the wind turbine wake region, which is characterised by large spatial gradients, the 10-minute averaged wind fields are used to estimate these gradients and execute a precision study on the effect of

a small pointing error on the actual measurement. Especially at the far end of the measurement domain, a small angular offset of 0.03° could cause a dislocation of the measurement point in the $y$-direction of 1 cm. When measuring a small scale effect in a wake, this displacement could affect the uncertainty significantly. In the following, we only consider the uncertainty in the $y$-direction, assuming this has the most significant contribution. Namely, the gradients in this direction are much steeper than in the $x$-direction, with an exception for the near vicinity of the rotor plane.

The uncertainty $e_y$ can be expressed in the azimuth angle pointing accuracy as follows:

$$e_y = \left( \sin\left( \chi + \frac{1}{2}e_\chi \right) - \sin\left( \chi - \frac{1}{2}e_\chi \right) \right) f \tag{7}$$

With this information, we are able to include an error on the $u$-component of the velocity according to the gradients in $y$-direction by means of the following uncertainty estimate:

$$e_{u_{wake}} = \frac{\partial u}{\partial y} e_y \tag{8}$$

The partial derivative $\frac{\partial u}{\partial y}$ is calculated numerically with a first order central finite difference coefficient based on the 10-minute averaged measurement itself (see Fig. 24a).

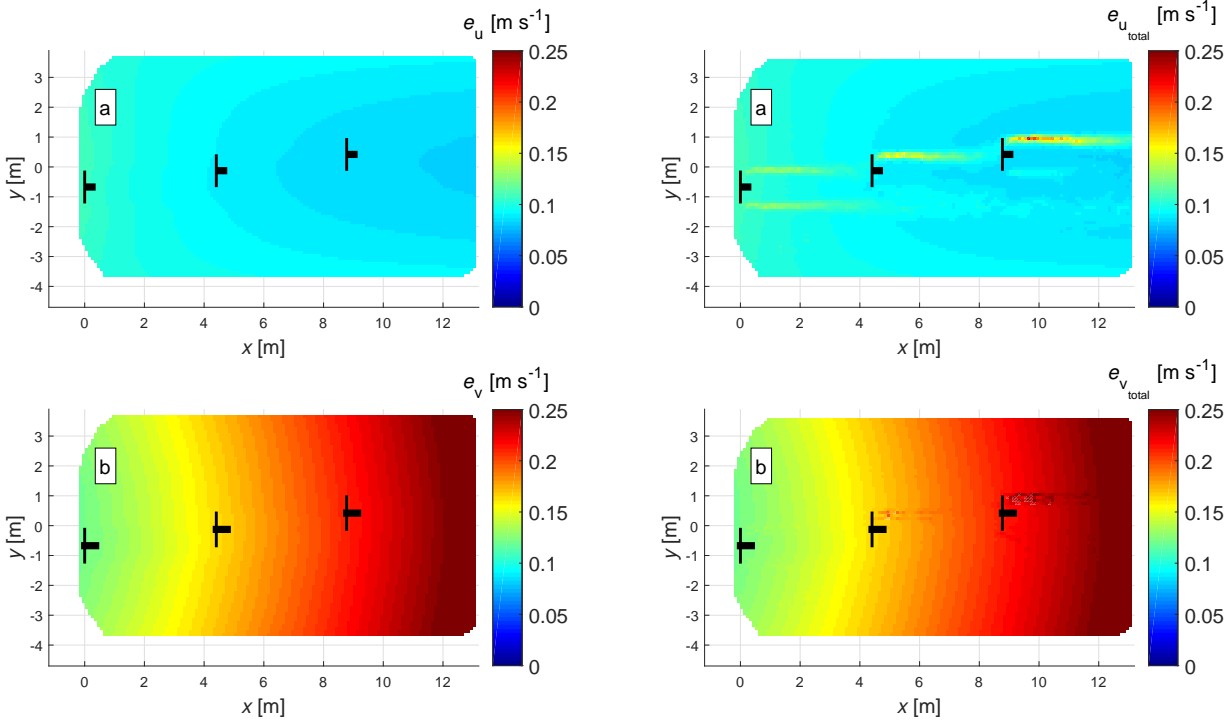

**Figure 25.** Dual-Doppler uncertainty on the evaluated $u$- and $v$-components; **a**: $u$, **b**: $v$.

**Figure 26.** Dual-Doppler uncertainty on the evaluated $u$- and $v$-components including wake gradient error; **a**: $u$, **b**: $v$.

Finally, the total uncertainty including the wake effect can be calculated as such:

$$e_{u_{total}} = \sqrt{e_u^2 + e_{u_{wake}}^2} \tag{9}$$

In Figs. 25a and 25b the 10-minute averaged uncertainty plots of both the evaluated $u$- and $v$-components, $e_u$ and $e_v$ respectively, are presented. It can be seen that the error $e_v$ is larger than $e_u$ overall. In the plot of $e_u$ it can be seen that the

error decreases slightly while moving from left to right. This is caused by the better alignment of the lidar beams with the $x$-direction. More interesting to analyse is the error on the $v$-component. It shows a significant increase towards the right of the measurement domain. This is related to the difference in azimuth angles between the two lidars, i.e. the lidar beams become more aligned with each other and thus have less potential to accurately resolve two orthogonal wind speed components. When the difference between the azimuth angles $|\Delta\chi| = |\chi_1 - \chi_2|$ tends towards $180°$, the $u$-component cannot be resolved anymore

and when $|\Delta\chi|$ tends towards $0°$, this applies to the $v$-component. To the far left of our measurement domain, $|\Delta\chi| \approx 90°$, which is the ideal case for dual-Doppler wind field reconstruction. To the far right of the plot, this angle difference decreases to $|\Delta\chi| \approx 30°$.

In Fig. 26 the respective total uncertainties $e_{u_{total}}$ are plotted, which include an error component related to the wake boundary gradients. As expected, an increased uncertainty around the boundaries is calculated, which gets larger with increasing distance

from the lidars. Particularly interesting is that on the lower part of the plots, the overlapping wakes from the three wind turbines smooth out the gradients, causing the error to be smaller than at the upper and steeper wake boundaries.

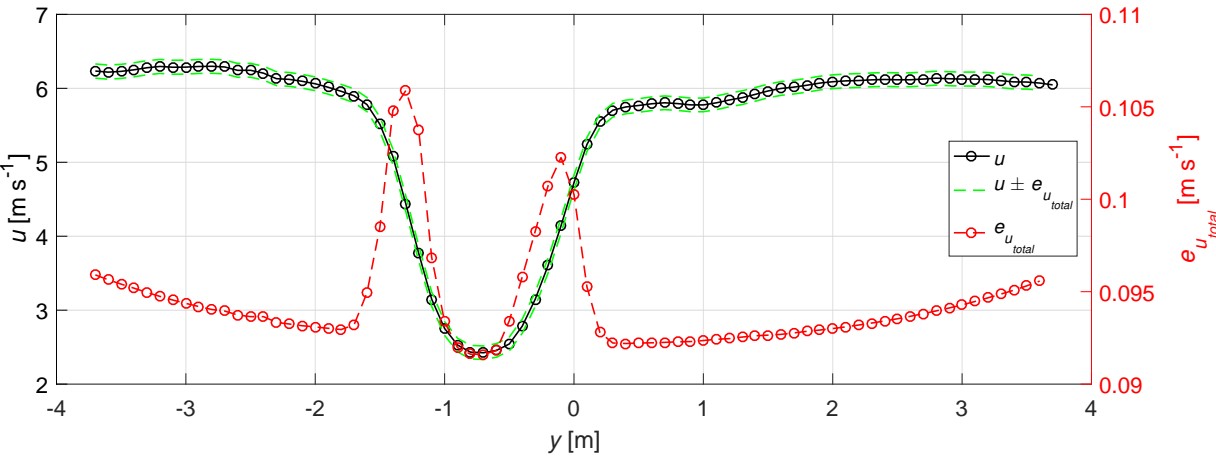

**Figure 27.** Average wake profile cross-section at a $2D$ downstream distance ($x = 2.2$ m) from the first turbine, with the total error indicated.

To get a better feeling for the magnitude of the additional error caused by the wake boundary gradients, Fig. 27 shows a wake profile at $2D$ downstream distance from turbine one, extracted from Fig. 26. Relative to the actual wake profile, the error bounds cannot be distinguished easily, but when observing the total error $e_{u_{total}}$ separately, an increase of up to 15% with

respect to the error $e_u$ can be noticed. This value will increase further when performing the same analysis further downstream.

This confirms that one has to be careful with lidar measurements in wind fields containing large gradients, even if the systems are known to have a high pointing accuracy.

## 4   Conclusions

A first measurement campaign with short-range synchronised WindScanner lidar measurements in a wind tunnel demonstrated that this technology can be used to measure both the wind tunnel mean flow and turbulence as well as wake profiles of scaled wind turbines. Validation was performed by comparing the lidar measurements with hot-wire probes, which yielded goodness of fit coefficients of 0.969 and 0.902 for the 1 Hz averaged $u$- and $v$-components of the wind speed, respectively. A downside is that the lidar systems cannot resolve the smallest turbulence scales, due to the finite measurement probe volumes which are significantly larger than those of the hot-wire probes. The true turbulence resolution the lidars provide is lower than their sampling frequency of 390 Hz, in this case even lower than 22 Hz. An extensive uncertainty analysis showed that increased errors occur in regions with steep spatial velocity gradients. However, lidar as a remote sensing application has the significant benefit that it does not influence the flow by its presence, contrary to the hot-wire probes, which have to be mounted on a beam structure that potentially disturbs the flow. Also, the WindScanner technology enables scanning and mapping of entire two-dimensional horizontal and vertical wind fields within seconds to minutes. It is therefore our conclusion that scanning wind lidars have significant potential for future wind tunnel measurement applications.

*Author contributions.*   M. F. van Dooren had the lead of both measurement analysis and paper writing. F. Campagnolo wrote the paragraph on the model wind turbines and provided further information about the measurement campaign. M. Sjöholm, N. Angelou and T. Mikkelsen designed the lidar measurement scenarios, performed the lidar measurement campaign, post-processed the lidar data, provided assistance in the lidar data analysis and interpretation, wrote parts of the content on the lidars, and critically reviewed the paper multiple times. M. Kühn initiated the measurement campaign, provided ideas for the scientific scope of the paper and had a supervising role.

*Competing interests.*   The authors declare that they have no conflict of interest.

*Acknowledgements.*   This work is partly funded by the German Ministry of Economic Affairs and Energy in the scope of the CompactWind project (Ref. Nr. 0325492B/D). Special thanks go out to all authors involved with the original work for the TORQUE 2016 conference: V. Petrović from the University of Oldenburg, C. L. Bottasso from the Technical University of Munich, and A. Croce and A. Zasso from the Politecnico di Milano. The authors also wish to thank all engineers and technicians who made this work possible: L. Ronchi, G. Campanardi, S. Giappino and D. Grassi from the Politecnico di Milano and P. Hansen and C. B. M. Pedersen from DTU Wind Energy.

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
