# Peer review of "Demonstration and uncertainty analysis of synchronised scanning lidar measurements of 2D velocity fields in a boundary-layer wind tunnel"

_Wind Energy Science, 2016_

## Referee Comment (RC1) · V. Iungo (Referee) · 6 Feb 2017

This manuscript deals with an assessment study of dual-Doppler measurements performed with two synchronized short-range continuous lidars against triple hot-wire data. 2D velocity measurements were performed at the wind tunnel of the Politecnico di Milano. The experimental campaign consisted of fixed point measurements, horizontal traverse in the wake of a wind turbine model and horizontal scan of three interacting wind turbine wakes.

The introduction briefly introduces to the topic under investigation, namely velocity

**WESD**

measurements of wind turbine wakes, both for field and wind tunnel measurements. In my opinion this section is a bit too short, and it could be expanded by providing an overview of the most representative studies for characterization of wind turbine wakes both for utility-scale wind turbines and down-scaled wind turbine models, which have been performed with different measurement techniques. In the introduction, I would also suggest to emphasize the importance of characterizing variability of wind turbine wakes with the daily cycle of the atmospheric static stability. In my opinion, this is a hot topic, which should be reproduced or mimicked through wind tunnel experiments as well. At the end of the introduction, I would add a paragraph stating clearly the aims of the project or the scientific questions that you aim to address with this research. This paragraph should be focused on the characterization of lidar capability to perform wind velocity measurements in a wind tunnel environment. Finally, I guess it will be convenient to add a description of the structure of the paper to facilitate reading.

A description of the wind tunnel, wind turbine model, lidars and hot-wire anemometer then follows. The different tests are presented in Sect. 2.5, while the results of the three different measurement strategies are reported in Sect. 3. An interesting uncertainty analysis is also provided in Sect. 3.4, and finally conclusions are reported in Sect. 4.

I commend the scientific contribution of this manuscript to the topics of characterization of wind turbine wakes and multiple-Doppler lidar measurements. I definitely recommend this manuscript for publication. However, in the following I report some minor comments, which might help to increase the thoroughness of the data analysis and impact on the wind energy community.

Comments:

1. P1 L21: I agree that a wind tunnel provides the great advantage of fixing different flow parameters, such as speed and turbulence intensity. However, I would mention that we should be able to reproduce the wind field variability consequent to the daily cycle of the atmospheric static stability, or mimic a realistic wind rose.

2. Sect. 1: I would suggest to expand this introduction to an overview of the most relevant works focused on characterization of wind turbine wakes both for utility-scale wind turbines and wind turbine models, which have been performed with different measurement techniques.

3. Sect 1: Add a paragraph to state clearly the aims of this project and the scientific questions you are attempting to address. I guess the main focus will be assessing the capabilities of continuous lidar to measure wind speed and turbulence in a wind tunnel environment.

4. Sect 1: Add at the end of the section a description of the structure of the paper.

5. P2 L15: Add some references for the wind turbine models and wind energy projects performed at the wind tunnel of PoliMi.

6. P3 L4: In my opinion, it is a bit too simplistic affirming that we can reproduce an ABL flow in a wind tunnel with spires and turbolators. We would need also to reproduce the temperature profile and typical length scales of the coherent structures. I would say that we can reproduce similar vertical profiles of wind speed and turbulence.

7. P3 L4: Please add a figure reporting the vertical profile of wind speed and turbulence intensity at the beginning of the turntable.

8. P4 L9: I would remove "averaged". Each measurement is a sample corresponding to a given sampling frequency.

9. P4 L11: Does this lidar have any blind region close to its location?

10. P4 L19: I think the setup and installation of the lidars in the wind tunnel is not described in detail. It would be interesting for the reader learning about the procedure you applied for the positioning and pointing of the two lidars.

11. P4 L27-28: In my opinion the elevation angle of 3° rather than a contamination from the w velocity, implies an under-estimation of the u and v velocities. It will be interesting to quantify error introduced on the horizontal velocities through the uncertainty analysis discussed later in Sect. 3.4.

12. P5 L9-13. Did you use the Dantec software for the calibration or do you have your own calibration procedure? In the second case, please provide a description and references.

13. P6 L5-11: It would have been interesting to estimate effects of the probe length on the turbulence statistics by varying the distance between the lidar and the measurement point. Do you have any available data to address this point?

14. P7 L1-3: The characteristics of the incoming velocity field should be reported in Sect. 2.1. Please provide a figure with the vertical profile of velocity, turbulence intensity and integral length scale of the wind tunnel flow.

15. P7 L6-10. It would be more effective to show a chunk of velocity signals for both anemometer and lidar showing the different filtering steps starting from the raw data. Something like Fig. 10, but starting from the initial sampling frequency of the two instruments.

16. Table 1: I would add to the statistics skewness and kurtosis in order to learn more from the statistical behavior of the two signals. Furthermore I would provide statistics for the raw signals (with their sampling frequency), and after 1-s averaging.

17. Fig. 16. I am not sure if you clarify a significant energy damping for frequencies lower than 28 Hz. I guess you should emphasize that the size of the virtual measurement volume might be much larger than 0.1 m and it's a function of the relative angle between the two-laser beams. A more detailed discussion

in this direction may help to better clarify this disagreement with the theoretical expectations.

---

## Referee Comment (RC2) · C. Westergaard (Referee) · 13 Mar 2017

No doubt the article should be published, it is a major contribution and clarification on the experiments performed. There are some confusing elements and also some technical structure which could be better, the comment will be focused one request of some clarification and improvements of the text.

Corrections needed:

C1 (major correction) Abstract, p1, l11 and l124 "the larger measurement probe volume" (L11) is directly in conflict with the actual results and the statement in L14 "lidars

to accurately measure small scale flow structures". This language should be more precise and accurate to the results. My suggestion would be to accurately describe two things: a) The measurement volume of the LIDARs is 13 to 50 cm, allowing average flow features to be resolved at 0.11D to 0.57D in the far downstream of the three turbine. b) The same volume limitation results in an effective spectral cut-off frequency between 22 and 11 Hz, respectively. This is compared to high resolution HW data.

C2 (minor), p1 L14 Why is "January 2016" important ? Maybe reformulate.

C3, (please elaborate) w-influence, p4, L24-28 w is neglected and deemed insignificant. Never the less, it is measured to be 0.08 m/s and must surely introduce a bias, especially as the vertical projection of the measurement volume is between 0.7 and 2.6 cm (at 3 deg downwards beam orientation, and the two ranges given in fig 5).

C3 (major correction) coordinates of fig 4 Please correct figure 4, so the coordinate system can be understood. Preferably x/D and Y/D, so it is comparable to figure 3

C4 (major correction) "rated conditions", P6, L15 and L19 (possible other places) "turbines operating under rated conditions" is nonsense. In rated conditions, the pitch of the blades is going positive and all research shows the wake disappears. So, I am pretty sure the author means that the turbine is operating close to maximum wake deficit or maximum Cp. Please be specific: TSR = xx, Pitch = xx, corresponding to an estimate Ct of xx, resulting an estimated wake deficit of xx.

C5 (minor) TI p7, L2 Why not be specific, TI=5.4% (as measured, table 1)

C6 (major) Table 1 and text on p7 There is an average v and w component. Albeit fairly small it is not really clear if this is a flow feature or probe misalignment ? Please be precise in the language about this.

C7 (major) Table 1 and text on p7 There is no difference between u' and v' in the table, but yet a very velar difference in figure 8 and 9 scatter ? Either observe or comment on this.

C8 w component Same comment as C3

C9 Error in figure 9 The offset in fig 9 red curve is indicated to be 0.00 – Looks like a mistake. Please check.

C10 (major) effects comparing fig 8/9 to 12/13 P9, L2 "the mentioned effects caused the scatter". I don't think this is accurate. The 3 mentioned effects (p7 L23 to p8 L2) is bias effects. Why would bias effects cause scatter ? The more likely effect is that the small scales are not resolved by the LIDAR.

C11 (minor) p9 L3 Delete "most of the very" And "are omitted here" maybe you mean is "averaged away by the large measurement volume"

C12 (minor) p9 L5 and L6 L5 "in the wind tunnel" should be "in the wind tunnel at 1 Hz". L6 "are estimated that well" should be "at 1 Hz follows the same trend".

C13 (major) p10, L10 The spatial resolution is given to be in the range 0.13 to 0.5 me. This gives $\frac{1}{2}$*5.6/0.13 to $\frac{1}{2}$*5.6/0.5, which is 22 Hz to 5.5 Hz temporal resolution.

It is generally misleading that it is suggested the LIDAR can measure up to 390 Hz. And the results shows the true range of resolution is 5.5 to 22 Hz. Please correct this throughout the paper.

C14 (major) figure 16 Introduce 5.5 to 22 Hz. 28 Hz is inaccurate. Also, 5/3 rule should be 5/3 Kolmogorov (since "rule" is not discussed)

C15 P11, figure 17/18 It would be prudent to present 22 Hz (or maybe 5.5Hz) filtered data.

C16 P11, comments pertaining to figure 18 and w-component It would be prudent to comment that at 3D the w-component is not insignificant and this could have resulted in the "v-component signature", otherwise the reader is left with the impression that this is the turbulence, which is probably not the case.

C17, P11, L16 (minor) "determine local 2D effects", I think you mean "measure a cross

section of a wind turbine wake" or something like that

C18, P11, L19 "as well as tip vortex". I don't think that is very clear. It could be turbulence in the shear layer, or the w-component or the shear layer itself.

C19, P11, L21 (major) I tin you either need to reference where the "turbulence in the lower region" has been observed in other studies, or reference shear layer or make it clear that your speculating. Because it could be due to velocity bias in the measurement.

C20 Figure 19 and 20 Introduce 19 a , 19 b and 20 a and 20 b for clarity.

C21 Figure 20 There is a clear periodicity in figure 20, v – component. Is this due to reconstruction or something else. The periodicity is about 1.8 m ? Also there is some strange signatures of u and v velocity at y/D = 1 and -3 which occurs to be artificial ?

C22 Figure 20 There is distinct different direction on the v-component coming of the rotor comparing turbine 1, 2 and 3. Why ? is there a missed observation ? measurement error due to w- component ?

C23 Figure 21 and 22 Introduce 21 a , 21 b and 22 a and 22 b for clarity.

C24, p12, section 3.4 I am not wildly convinced the uncertainty analysis is fully representative. The two conditions given on pg. 12 does not include a number of factors. Also there is not consideration of bias errors, which may very well be higher than the uncertainty presented. I appreciate the authors can not give this now, but at least elaborate a bit more on other potential sources and their nature.

C25 Conclusion This: "Because of the lidar measurement principle, between 10-15% of the data is lost due to the moving wind turbine blades in the measurement region" is hardly discussed in the paper, and not what the paper is about. Suggest to strike. It would be prudent to summarize resolution results, frequency and measurement volume overD. Also, comment on future work ?

Please also note the supplement to this comment:
http://www.wind-energ-sci-discuss.net/wes-2016-59/wes-2016-59-RC2-supplement.pdf

---

## Referee Comment (RC3) · Anonymous Referee #3 · 14 Mar 2017

The paper combines hot-wire and lidar measurements in a wind tunnel. Main focus is on the uncertainty analysis. The experiments seem to be carefully designed and executed. The results are interesting for a large audience. I recommend publication of the paper in the Wind Energ. Sci after following mostly minor issues are addressed.

How did the authors calibrate 3D hot-wire probe? How about accuracy of the calibration? Is there any particular reason for choosing a 3D hot-wire probe even though the lidar measurements were 2D?

The authors worry about heating of the hot-wire probe. There have been many studies

in literature where LDA and hot-wire or PIV and hot-wire were used in order to measure different flows both in and outside of wind tunnels. Why in this case the heating caused by the laser beam becomes an issue.

The authors carried out the measurements of three different types: In the second case, the complete line was covered every 1 s with equally sampled measurements. Here, the characteristics scales of turbulence should be given for quantitative comparison. For example, what is the corresponding integral scale for this 1 s measurements. Is 1 sec statement correct?

Right before the results the authors give information about the wind velocity profile. Since the work compares the two system, it is of interest to see the velocity and turbulence profiles upstream of the turbines. These profiles should also be compared again the theoretical profiles like the power law instead of giving turbulence intensity at one single location.

It is not clear why they carried out the measurements at 2500 Hz for hot-wire and 390 Hz for lidar and then compared at 1 Hz. Any particular reasoning for carrying out the measurements this way?

It is not clear if the figures 8 and 9 are based on instantaneous readings or the mean quantities, or turbulence. The authors call the figures correlation, but as far as seen in these figures they are individual (or mean) points from one instrument and corresponding reading from the other instrument. How accurate the time lag was introduced into these computations? Also it is difficult to figure out the motivation for red fits in these figures, 8 and 9. Is there any particular significance? It would also be nice to see how they compute what is presented in figures 8 and 9.

Figure 10 and 11 show very good agreement between these two signals. Can the authors elaborate effect of down sampling from higher sampling rates on the line plots?

Discussion regarding the figure 16 needs to be further detailed. For example what is

the record length for these spectral computations, and how many blocks of data are used. Even though the Taylors theorem indicates 28 Hz, lidar seems to be rolling off much earlier around at 10 Hz. Any explanation for this? In addition, the authors should write the formulation used for computing spatial averaging when finding 28 Hz. The authors should also write how they computed the spectra.

Figure 17 and 18: Here it is interesting to know about how many effective uncorrelated samples in these 1 minute recordings across the wake. Statistical accuracy changes for a fixed record length since the turbulence intensity vary a lot. What do the author mean by binned average? is this average of the open circles?

The authors note that it is hard to give any hard conclusion on the lidar's ability to measure small scale turbulent fluctuations. Previously the authors showed this when presenting the spectra. According to their approximation the cut-off frequency is about 28 Hz, which is rather low considering the probable length of the cascade, which is hard to find out since the Reynolds number is not stated as far as seen.

On page 13 and in line 15, the authors writes about the small scale effect such as wake. What do they mean here? I think it is not possible to capture the small scales using this setup due to the size of measurement volume, and wake itself cannot be considered small scale.

From the formulation given in the text, it is not obvious that the uncertainty in the y-direction has the most significant contribution. Further explanation is needed for this statement.

When the author discuss about the total uncertainty, they mostly relate it to angle difference. Due to the nature of the flow, however, turbulence intensity also plays an important role in any uncertainty computation due to the statistical convergence. Toward the edges of the wake, the mean velocity drops and turbulent fluctuations as well. But the intensity can be very high? What would be the effect of this on their uncertainty calculations. Another question in relation to this one is that the wake develops downstream and velocity deficit gets smaller and smaller, and this leads to stronger demand on resolution. What would be the effect of this on the performance of lidar data, and the uncertainty. One can look at figure 21 and 22 to get an idea, but there the uncertainty is higher along the tip vortex, and wake development does not matter.
* * *

---

## Author Comment (AC1) · 11 Apr 2017

Dear Dr. Iungo,

Thank you very much for the critical and in-depth review of our manuscript. It has an overall positive tone, but convincingly addresses some shortcomings of the paper that can be improved to prepare it for publication in the Wind Energy Science journal. Below we will address each of your comments separately.

1. P1 L21: I agree that a wind tunnel provides the great advantage of fixing different flow parameters, such as speed and turbulence intensity. However, I would mention that we should be able to reproduce the wind field variability consequent to the daily cycle of the atmospheric static stability, or mimic a realistic wind rose.

   It is indeed good to mention this drawback of measuring in a wind tunnel. Therefore we added the sentence ‚*Please note that one of the shortcomings of measuring in a wind tunnel as opposed to free-field measurements is the ability of simulating the variability of atmospheric stability and a representative wind rose.*'.

2. Sect. 1: I would suggest to expand this introduction to an overview of the most relevant works focused on characterization of wind turbine wakes both for utility scale wind turbines and wind turbine models, which have been performed with different measurement techniques.

   We will add some references that give an overview of the state-of-the-art research being done on wind turbine wake characterisation in both the free-field and in wind tunnel experiments. However, we would like to stress that the objective of this paper is not to do an extensive wake analysis, but it uses wake measurement to provide an interesting and relevant example. This objective will be highlighted more clearly (see your **comment #3**).

3. Sect 1: Add a paragraph to state clearly the aims of this project and the scientific questions you are attempting to address. I guess the main focus will be assessing the capabilities of continuous lidar to measure wind speed and turbulence in a wind tunnel environment.

   You are right. At the end of the introduction, we now added the objective as the sentence ‚*The objective is to assess the capabilities of continuous-wave short-range lidar to map wind flows and measure turbulence in a wind tunnel.*'.

4. Sect 1: Add at the end of the section a description of the structure of the paper.

   The structure of the paper will be included at the end of the introduction.

5. P2 L15: Add some references for the wind turbine models and wind energy projects performed at the wind tunnel of PoliMi.

   We believe that there are sufficient references included that are related to both the wind tunnel as well as the model wind turbines. However, they were scattered over the paper and are now repeated in the section you mentioned to have a better consistency for reading.

6. P3 L4: In my opinion, it is a bit too simplistic affirming that we can reproduce an ABL flow in a wind tunnel with spires and turbolators. We would need also to reproduce the temperature profile and typical length scales of the coherent structures. I would say that we can reproduce similar vertical profiles of wind speed and turbulence.

   This is true, the sentence might have exaggerated what the turbulence generators are capable of. We changed it to ‚*Typical vertical profiles of wind speed and turbulence can be imitated by the use*

*of bricks on the floor that act as roughness and turbulence generators, i.e. spires, placed at the chamber inlet at the left boundary.'*.

7.  P3 L4: Please add a figure reporting the vertical profile of wind speed and turbulence intensity at the beginning of the turntable.

    Unfortunately we did not execute measurements of the vertical profile during this measurement campaign. However, vertical wind profiles for both wind speed and turbulence were measured during a previous campaign, under identical conditions. These plots are added to the manuscript.

8.  P4 L9: I would remove "averaged". Each measurement is a sample corresponding to a given sampling frequency.

    The word *,averaged*' was replaced by *,Doppler spectrum averaged*' to indicate that it is an average over the whole time interval and not just a snap-shot, i.e. the raw signal is actually sampled at 100 MHz, Fourier transformed, and then averaged.

9.  P4 L11: Does this lidar have any blind region close to its location?

    There is no blind region close to the lidar, but the optical configuration of the device implies a minimum and maximum possible focus distance. To improve the clarity, the sentence was changed to *,The measurement range is defined by the optical configuration of the device, which enables motor controlled focus distance between about 9 m and 150 m.*'.

10. P4 L19: I think the setup and installation of the lidars in the wind tunnel is not described in detail. It would be interesting for the reader learning about the procedure you applied for the positioning and pointing of the two lidars.

    The procedure for installation and calibration of the lidar has now been extended significantly, in order to have a more complete explanation.

11. P4 L27-28: In my opinion the elevation angle of 3° rather than a contamination from the w velocity, implies an under-estimation of the u- and v-velocities. It will be interesting to quantify error introduced on the horizontal velocities through the uncertainty analysis discussed later in Sect. 3.4.

    Indeed this would be very interesting and other reviewers also mentioned the possible bias in the horizontal wind speed components caused by neglecting the vertical wind component. This will be investigated further by means of uncertainty analysis.

12. P5 L9-13. Did you use the Dantec software for the calibration or do you have your own calibration procedure? In the second case, please provide a description and references.

    The Dantec hot-wire probes were calibrated according to their factory software and manual. The procedure is now explained in detail and a reference is provided.

13. P6 L5-11: It would have been interesting to estimate effects of the probe length on the turbulence statistics by varying the distance between the lidar and the measurement point. Do you have any available data to address this point?

    This would be very interesting indeed. We admit that there is a high dependency on the capability of measuring small-scale turbulence and the probe length, the latter of which increases quadratically with the focus distance. However at this point there is no such data set with which we can assess this issue.

14. P7 L1-3: The characteristics of the incoming velocity field should be reported in Sect. 2.1. Please provide a figure with the vertical profile of velocity, turbulence intensity and integral length scale of the wind tunnel flow.

We now shifted the information about the wind tunnel inflow velocity and turbulence intensity to section 2.1. Also, as we answered to your **comment #7**, we included vertical profiles of wind speed and turbulence from a previous campaign executed under identical conditions and wind tunnel settings.

15. P7 L6-10. It would be more effective to show a chunk of velocity signals for both anemometer and lidar showing the different filtering steps starting from the raw data. Something like Fig. 10, but starting from the initial sampling frequency of the two instruments.

We now included the time series for the 390 Hz $u$- and $v$-components as well, before averaging them to 1 Hz time series.

16. Table 1: I would add to the statistics skewness and kurtosis in order to learn more from the statistical behavior of the two signals. Furthermore I would provide statistics for the raw signals (with their sampling frequency), and after 1-s averaging.

The statistical parameters skewness and kurtosis have been added to the table. Also an additional table with the statistics of the 1 Hz averaged time series has been included in addition.

17. Fig. 16. I am not sure if you clarify a significant energy damping for frequencies lower than 28 Hz. I guess you should emphasize that the size of the virtual measurement volume might be much larger than 0.1 m and it's a function of the relative angle between the two-laser beams. A more detailed discussion in this direction may help to better clarify this disagreement with the theoretical expectations.

We clarified the observed behaviour with the text ,*The drop in the slope of the spectrum does not exactly coincide with the 28 Hz frequency mark, because the intrinsic Lorentzian spatial weighting function of a continuous-wave lidar extends beyond the defined bounds of the probe length, therefore also acting as a filter on lower frequencies. The effect of spatial weighting is explained in detail by Sjöholm et al. (2009). Also combining measurements from two lidars that each have a different probe volume causes an even larger effect of averaging out small turbulence scales over a more complex x-shaped volume (see Fig. 4).'.* We believe that this should be sufficient to understand that frequencies lower than 28 Hz are already affected by the volume averaging effect.

---

## Author Comment (AC2) · 11 Apr 2017

Dear Dr. Westergaard,

We appreciate your recommendation for this article to be published. Also thank you for the high level of detail in your review, which we will take into account very seriously. Below we will address each of your comments separately.

1. (major correction) Abstract, p1, l11 and l124 "the larger measurement probe volume" (L11) is directly in conflict with the actual results and the statement in L14 "lidars to accurately measure small scale flow structures". This language should be more precise and accurate to the results. My suggestion would be to accurately describe two things: a) The measurement volume of the LIDARs is 13 to 50 cm, allowing average flow features to be resolved at 0.11D to 0.57D in the far downstream of the three turbine. b) The same volume limitation results in an effective spectral cut-off frequency between 22 and 11 Hz, respectively. This is compared to high resolution HW data.

   It is true that there is a conflict between the lidar's ability to measure small scale flow stuctures on the one hand, and the probe length averaging on the other. Therefore this will be rewritten to provide a better picture of the capabilities. Your concerns regarding the measurement volumes will be addressed in your **comment #13**.

2. (minor), p1 L14 Why is "January 2016" important ? Maybe reformulate.

   We thought it is relevant side information to know when the measurement campaign took place. However, we will move this date to the end of the sentence, where it fits better with regard to its importance.

3. (please elaborate) w-influence, p4, L24-28 w is neglected and deemed insignificant. Never the less, it is measured to be 0.08 m/s and must surely introduce a bias, especially as the vertical projection of the measurement volume is between 0.7 and 2.6 cm (at 3 deg downwards beam orientation, and the two ranges given in fig 5).

   It is a good observation that the vertical wind speed component indeed has a contribution and one could argue that it is not insignificant. Therefore we will extend the uncertainty analysis by incorporating the potential bias in the other two wind speed components that could be caused by neglecting the vertical component.

   (major correction) coordinates of fig 4 Please correct figure 4, so the coordinate system can be understood. Preferably x/D and Y/D, so it is comparable to figure 3

   The only purpose of Figure 4 is actually to sketch the shape of the probe volumes and therefore it is not plotted to any scale. If we use the scale that corresponds to Figure 3, one can barely see the probe volumes, since their dimensions are so much smaller than the focus distances. Therefore we prefer to keep it like this.

4. (major correction) "rated conditions", P6, L15 and L19 (possible other places) "turbines operating under rated conditions" is nonsense. In rated conditions, the pitch of the blades is going positive and all research shows the wake disappears. So, I am pretty sure the author means that the turbine is operating close to maximum wake deficit or maximum Cp. Please be specific: TSR = xx, Pitch = xx, corresponding to an estimate Ct of xx, resulting an estimated wake deficit of xx.

According to our point of view ‚rated conditions' occur exactly at the one point where the wind turbine first reaches maximum power (also maximum $C_p$ and $C_T$ are theoretically achieved here) and the transition between torque control and pitch control takes place. At this point you indeed have the highest wake deficit. The pitching will occur for wind speeds between the rated wind speed and the cut-out wind speed, in which case the wake is indeed expected to disappear. We will make sure that the reader knows which operational conditions we mean. Therefore we will also provide values for the tip speed ratio, pitch angle and the thrust coefficient of the wind turbine during each part of the experiment.

5. (minor) TI p7, L2 Why not be specific, TI=5.4% (as measured, table 1)

Good point to state the more precise measured turbulence intensity here. This will be adjusted.

6. (major) Table 1 and text on p7 There is an average v and w component. Albeit fairly small it is not really clear if this is a flow feature or probe misalignment ? Please be precise in the language about this.

At the time of writing this response, it is not known where this bias stems from, but it will definitely be investigated and we agree that this has to be addressed in the revised manuscript.

7. (major) Table 1 and text on p7 There is no difference between u' and v' in the table, but yet a very velar difference in figure 8 and 9 scatter ? Either observe or comment on this.

This is indeed an interesting observation which we did not pay much attention to so far. It could just be caused by the scaling of the plot and the range of the wind speeds, but this will be checked.

8. w component Same comment as C3

See the answer on **comment #3**.

9. Error in figure 9 The offset in fig 9 red curve is indicated to be 0.00 – Looks like a mistake. Please check.

In Figure 9 it can be clearly seen that the red regression line crosses the point (0,0), so we could not identify what error you mean. However, it is strange that the slope is quite far off unity. This will be reassessed.

10. (major) effects comparing fig 8/9 to 12/13 P9, L2 "the mentioned effects caused the scatter". I don't think this is accurate. The 3 mentioned effects (p7 L23 to p8 L2) is bias effects. Why would bias effects cause scatter ? The more likely effect is that the small scales are not resolved by the LIDAR.

True, after looking through the reasons given for the scatter, it is concluded that only the first point could indeed be valid. However the other two points could be an explanation for a bias but not for scatter. Also the lidar not being able to resolve small scales is a possible reason. Therefore this part will be slightly rewritten according to this concern.

11. (minor) p9 L3 Delete "most of the very" And "are omitted here" maybe you mean is "averaged away by the large measurement volume"

You are right that this sentence is not very accurate. Therefore we changed it to '...*since small scale fluctuations are averaged out*.'.

12. (minor) p9 L5 and L6 L5 "in the wind tunnel" should be "in the wind tunnel at 1 Hz". L6 "are estimated that well" should be "at 1 Hz follows the same trend".

It is indeed good to state that this validation is only valid for the 1 Hz measurements. That is why we adapted your suggestion here.

13. (major) p10, L10 The spatial resolution is given to be in the range 0.13 to 0.5 me. This gives 1/ 2*5.6/0.13 to 1/2 *5.6/0.5, which is 22 Hz to 5.5 Hz temporal resolution.

It is good that you mention this, because there is a slight inconsistency in the manuscript. Namely the probe lengths of 0.13 m and 0.50 m indicated in Figure 5 are valid for focus distances of 10 and 20 m, respectively. However the point measurements were actually executed at a focus distance of 9 m, yielding a probe length of 0.10 m instead of 0.13 m and this translates to the calculated 28 Hz resolution. We will make sure make this consistent throughout the manuscript (in the text and the figures).

It is generally misleading that it is suggested the LIDAR can measure up to 390 Hz.
And the results shows the true range of resolution is 5.5 to 22 Hz. Please correct this throughout the paper.

It is also true that, although the lidar has a sampling rate of 390 Hz, the true resolution of small scales is actually lower. This difference will be described more clearly, also taking into consideration your **comment #1**.

14. (major) figure 16 Introduce 5.5 to 22 Hz. 28 Hz is inaccurate. Also, 5/3 rule should be 5/3 Kolmogorov (since "rule" is not discussed)

Please find the reason for using 28 Hz in **comment #13**. The manuscript will be adjusted to be consistent. Also the Komolgorov rule will be introduced in Figure 16 as you suggest.

15. P11, figure 17/18 It would be prudent to present 22 Hz (or maybe 5.5Hz) filtered data.

In the case of Figure 16, we do not see the need to post-process the data by filtering it to 22 Hz.

16. P11, comments pertaining to figure 18 and w-component It would be prudent to comment that at 3D the w-component is not insignificant and this could have resulted in the "v-component signature", otherwise the reader is left with the impression that this is the turbulence, which is probably not the case.

You raised the valid issue here that the vertical wind speed component ($w$) might have a significant effect here and even causes the signature in the lateral ($v$-)component of the flow. This will be discussed in the paper. Also see the answer on your **comment #3**.

17. P11, L16 (minor) "determine local 2D effects", I think you mean "measure a cross section of a wind turbine wake" or something like that

Since this statement was indeed a bit vague, we changed this sentence to ,*It (the figure) illustrates that the lidars are capable of determining the two-dimensional flow across a wind turbine wake cross-section.'*.

18. P11, L19 "as well as tip vortex". I don't think that is very clear. It could be turbulence in the shear layer, or the w-component or the shear layer itself.

It will be discussed in the paper that tip vortices cannot be identified with certainty, and the possible other reasons you give will be mentioned and discussed.

19. P11, L21 (major) I tin you either need to reference where the "turbulence in the lower region" has been observed in other studies, or reference shear layer or make it clear that your speculating. Because it could be due to velocity bias in the measurement.

Let it be clear that the ,lower region' refers to the lower part of the plot. The wording might be a bit off here, but we are quite certain that some local effects in the v-component are ,washed away' by the presence of the other overlapping wakes. We will try to state this in a different way and make sure that this is a speculation and not a hard fact.

20. Figure 19 and 20 Introduce 19 a , 19 b and 20 a and 20 b for clarity.

All subfigures will be alphabetically numbered. See also **comment #23**.

21. Figure 20 There is a clear periodicity in figure 20, v – component. Is this due to reconstruction or something else. The periodicity is about 1.8 m ? Also there is some strange signatures of u and v velocity at y/D = 1 and -3 which occurs to be artificial ?

This periodicity is most likely caused by the wind field reconstruction method or interpolation scheme. It will be investigated whether these effects can be filtered out. Also the ,strange signature' is probably caused by invalid measurements. They will be either filtered out or their presence will be discussed.

22. Figure 20 There is distinct different direction on the v-component coming of the rotor comparing turbine 1, 2 and 3. Why ? is there a missed observation ? measurement error due to w-component?

We think that directional differences are mainly caused by the interaction of the multiple wakes. However, this lies beyond the scope of the paper, so it will not be discussed.

23. Figure 21 and 22 Introduce 21 a , 21 b and 22 a and 22 b for clarity.

All subfigures will be alphabetically numbered. See also **comment #20**.

24. p12, section 3.4 I am not wildly convinced the uncertainty analysis is fully representative. The two conditions given on pg. 12 does not include a number of factors. Also there is not consideration of bias errors, which may very well be higher than the uncertainty presented. I appreciate the authors can not give this now, but at least elaborate a bit more on other potential sources and their nature.

It is true that the uncertainty analysis is not representative of all physical effects that possibly have an influence. It is mainly aimed at modeling how a given uncertainty or bias is propagated through the dual-Doppler lidar reconstruction. However, in this way of expressing the error, there is no clear separation between bias and uncertainty. A bias in either the line-of-sight measurement or one of the scanning angles will be propagated in the exact same way. Potential sources for uncertainty and biases will be elaborated on further.

25. Conclusion This: "Because of the lidar measurement principle, between 10-15% of the data is lost due to the moving wind turbine blades in the measurement region" is hardly discussed in the paper, and not what the paper is about. Suggest to strike. It would be prudent to summarize resolution results, frequency and measurement volume over D. Also, comment on future work ?

You are right that the conclusion should not give this new information, without it being mentioned anywhere else. Also the sentence is not clear. Therefore this sentence will be omitted and the data availability will be mentioned somewhere earlier in the paper. Also you stress the importance of concluding on the lidar resolution study, which is valid and will be emphasised in the conclusion in a better way.

---

## Author Comment (AC3) · 11 Apr 2017

Dear anonymous referee,

Thank you very much for your review and your recommendation for publishing this work after a minor revision. We will address all of your comments, which we have numbered 1-13, separately in the following.

1. How did the authors calibrate 3D hot-wire probe? How about accuracy of the calibration? Is there any particular reason for choosing a 3D hot-wire probe even though the lidar measurements were 2D?

   The calibration and accuracy of the hot-wire probes will be stated in the paper more thoroughly. The reason for choosing this device is simply that it was part of the standard setup in the wind tunnel. Also based on the other reviews we received, we will pay more attention to the missing vertical wind speed ($w$-) component. By using the information of the hot-wire we will be able to address the uncertainty it adds to the reconstructed $u$- and $v$-components of the lidar.

2. The authors worry about heating of the hot-wire probe. There have been many studies in literature where LDA and hot-wire or PIV and hot-wire were used in order to measure different flows both in and outside of wind tunnels. Why in this case the heating caused by the laser beam becomes an issue.

   Because we did not know whether or not it could have an influence, we just made sure that the laser beam could not hit the hot-wire itself, as a preventive measure. We believe it is a valid speculation that a laser beam hitting a very sensitive metal wire could interact with it and therefore we focused the beam a small distance away from the hot-wire probe.

3. The authors carried out the measurements of three different types: In the second case, the complete line was covered every 1 s with equally sampled measurements. Here, the characteristics scales of turbulence should be given for quantitative comparison. For example, what is the corresponding integral scale for this 1 s measurements. Is 1 sec statement correct?

   The turbulence scales that can effectively be measured with this setup are larger than the ones corresponding to 1 Hz, because the temporal and spatial resolution are linked. This will be mentioned in the paper.

4. Right before the results the authors give information about the wind velocity profile. Since the work compares the two system, it is of interest to see the velocity and turbulence profiles upstream of the turbines. These profiles should also be compared again the theoretical profiles like the power law instead of giving turbulence intensity at one single location.

   Unfortunately we did not execute measurements of the vertical profile during this measurement campaign. However, a vertical wind profile for both wind speed and turbulence were measured during a previous measurement campaign, under identical conditions. These plots are added to the manuscript.

5. It is not clear why they carried out the measurements at 2500 Hz for hot-wire and 390 Hz for lidar and then compared at 1 Hz. Any particular reasoning for carrying out the measurements this way?

We measured with both devices at their maximum sampling rates, to benefit the most from their turbulence measurements. However, since 2500 Hz and 390 Hz are not compatible, we first averaged the hot-wire measurements to 390 Hz to allow for a fair comparison. But because we know that the lidar probe volume averaging effect filters out small turbulence scales and does not truly resolve turbulence scales up to 390 Hz, also a comparison at a more reasonable 1 Hz was done additionally.

6.  It is not clear if the figures 8 and 9 are based on instantaneous readings or the mean quantities, or turbulence. The authors call the figures correlation, but as far as seen in these figures they are individual (or mean) points from one instrument and corresponding reading from the other instrument. How accurate the time lag was introduced intothese computations? Also it is difficult to figure out the motivation for red fits in these figures, 8 and 9. Is there any particular significance? It would also be nice to see how they compute what is presented in figures 8 and 9.

Figures 8 and 9 are regression curves of the wind speed components $u$ and $v$, showing how well the lidar and the hot-wire probe correlate with each other. We believe that it is standard practice to plot this and fit a regression curve through the scattered points (see the red line). We don't understand what more motivation we should provide for doing so. The plots are based on the instantaneous lidar measurements (390 Hz) and the averaged (from 2500 to 390 Hz) hot-wire measurements. The time lag was computed numerically based on a cross-correlation function, which finds the maximum correlation as a function of time lag. The accuracy of this method is the time step in between the lidar measurements, i.e. 1/390 Hz ≈ 2.6 ms.

7.  Figure 10 and 11 show very good agreement between these two signals. Can the authors elaborate effect of down sampling from higher sampling rates on the line plots?

The lidars filter out some small scale turbulence because of the probe length averaging effect. This means that the 390 Hz regression plots are not representative of the capabilities of the lidars. When averaging to 1 Hz, all turbulent scales can be measured as well by the lidar as by the hot-wire probe and the fit becomes better. To investigate this effect further, the plots in Figures 14 and 15 were produced.

8.  Discussion regarding the figure 16 needs to be further detailed. For example what is the record length for these spectral computations, and how many blocks of data are used. Even though the Taylors theorem indicates 28 Hz, lidar seems to be rolling off much earlier around at 10 Hz. Any explanation for this? In addition, the authors should write the formulation used for computing spatial averaging when finding 28 Hz. The authors should also write how they computed the spectra.

The spectrum is based on a two minute time series of the 390 Hz data, which is split into ten blocks which are then filtered with a Hann window to smooth the spectrum. This information will be added. The fact that the slope of the lidar spectrum already deviates from the -5/3 Komolgorov rule earlier on is explained with the sentence ,*The drop in the slope of the spectrum does not exactly coincide with the 28 Hz frequency mark, because the intrinsic Lorentzian spatial weighting function of a continuous-wave lidar extends beyond the defined bounds of the probe length, therefore also acting as a filter on lower frequencies.*'. This could be reformulated to increase the clarity.

9.  Figure 17 and 18: Here it is interesting to know about how many effective uncorrelated samples in these 1 minute recordings across the wake. Statistical accuracy changes for a fixed record length since the turbulence intensity vary a lot. What do the author mean by binned average? is this average of the open circles?

The red and green lines indeed show the mean and the standard deviation of the measurements marked with grey circles, binned with respect to the $y/D$-axis. Naturally there is a high uncertainty in the measurements, because the samples might not all be correlated. Therefore the mean profile here is more relevant than the single measurements.

10. The authors note that it is hard to give any hard conclusion on the lidar's ability to measure small scale turbulent fluctuations. Previously the authors showed this when presenting the spectra. According to their approximation the cut-off frequency is about 28 Hz, which is rather low considering the probable length of the cascade, which is hard to find out since the Reynolds number is not stated as far as seen.

We don't have hard numbers for the expected frequency range of the inertial subrange in the case of these wind tunnel measurements. However, the spectrum of the hot-wire probe measurements indicates that it extends roughly from 2 Hz to over 100 Hz, something the lidar measurements are unable to resolve. This difference between the two measurement devices is the most important point here.

You are also indicating that we actually are able to give a quantitative measure of the lidar's ability to measure turbulent fluctuations, namely the cut-off frequency of about 10 Hz (lower than the expected 28 Hz). We will reformulate this accordingly.

11. On page 13 and in line 15, the authors writes about the small scale effect such as wake. What do they mean here? I think it is not possible to capture the small scales using this setup due to the size of measurement volume, and wake itself cannot be considered small scale.

It is true that the wake is not a small scale itself. We meant that normally wakes introduce small scale effects such as a higher turbulence and shear on its boundary. Also we were not able to capture the really small scales in this setup. Therefore the text about this will be rewritten.

12. From the formulation given in the text, it is not obvious that the uncertainty in the y-direction has the most significant contribution. Further explanation is needed for this statement.

The fact that only the gradient $\frac{\delta u}{\delta y}$ was taken into account when assessing the uncertainty introduced by the wake being present, is that the gradient $\frac{\delta u}{\delta x}$ is very small almost everywhere in the wind field. The reason for this is that the wake recovery takes place very gradually, but the boundary of the wake shows very steep gradients along the $y$-direction. This will be written more clearly in the paper for a better understanding.

13. When the author discuss about the total uncertainty, they mostly relate it to angle difference. Due to the nature of the flow, however, turbulence intensity also plays an important role in any uncertainty computation due to the statistical convergence. Toward the edges of the wake, the mean velocity drops and turbulent fluctuations as well. But the intensity can be very high? What would be the effect of this on their uncertainty calculations. Another question in relation to this one is that the wake develops downstream and velocity deficit gets smaller and smaller, and this leads to stronger demand on resolution. What would be the effect of this on the performance of lidar data, and the uncertainty. One can look at figure 21 and 22 to get an idea, but there the uncertainty is higher along the tip vortex, and wake development does not matter.

It is true that the uncertainty analysis is mostly focusing on the lidar setup, i.e. the difference in the azimuth angle of the two lidars. The emphasis of the uncertainty analysis is the error introduced on the reconstructed wind speed components by the dual-Doppler lidar reconstruction. We do not have reliable measurements of how the turbulence intensity varies over the wind field and in the complex case of the wake, this is surely an important issue. However, this is not regarded here because it is out of the scope of our paper. Therefore a simplified model was established to express the uncertainty added by the wake's deterministic properties.

As stated before, the uncertainty of the dual-Doppler reconstruction is the most important aspect considered here. The wake development does not have a direct effect on the **absolute** value of the uncertainty. If you express it as a percentage of the velocity deficit, the uncertainty will grow, but according to us this is an artificial effect. Therefore we chose to not express the uncertainty in terms of the wake deficit.